# Potential of *Staphylea holocarpa* Wood for Renewable Bioenergy

**DOI:** 10.3390/molecules28010299

**Published:** 2022-12-30

**Authors:** Yiyang Li, Erdong Liu, Haiping Gu, Junwei Lou, Yafeng Yang, Longhai Ban, Wanxi Peng, Shengbo Ge

**Affiliations:** 1School of Forestry, Henan Agricultural University, Zhengzhou 450002, China; 2Department of Agricultural and Forestry Sciences, Henan Zhumadian Agricultural School, Zhumadian 463000, China; 3School of Architectural Engineering, Zhejiang Business Technology Institute, Ningbo 315012, China; 4Office of Academic Research, State Owned Madao Forest Farm of Biyang County, Zhumadian 463000, China; 5Jiangsu Co-Innovation Center of Efficient Processing and Utilization of Forest Resources, College of Materials Science and Engineering, Nanjing Forestry University, Nanjing 210037, China

**Keywords:** *Staphylea holocarpa*, lignocellulose, GC–MS, Py/GC–MS, NMR

## Abstract

Energy is indispensable in human life and social development, but this has led to an overconsumption of non-renewable energy. Sustainable energy is needed to maintain the global energy balance. Lignocellulose from agriculture or forestry is often discarded or directly incinerated. It is abundantly available to be discovered and studied as a biomass energy source. Therefore, this research uses *Staphylea holocarpa* wood as feedstock to evaluate its potential as energy source. We characterized *Staphylea holocarpa* wood by utilizing FT–IR, GC–MS, TGA, Py/GC–MS and NMR. The results showed that *Staphylea holocarpa* wood contained a large amount of oxygenated volatiles, indicating that it has the ability to act as biomass energy sources which can achieve green chemistry and sustainable development.

## 1. Introduction

Human civilization has reached a new historical height with current developments in science and technology, but it has also increased its utilization of energy simultaneously. Energy is extremely important for the development of the material basis and society, and is particularly important economically [1]. As an important material base, energy has played an indispensable role in nation building and human development. These include GDP growth, social and scientific development and the security of the nation [2]. At present, most countries in the world are still obtaining most of their energy from fossil energy [3]. However, the drastic reduction in fossil energy and the ecological environment has brought many problems. Therefore, it is necessary to develop renewable energy in order to alleviate the over-consumption of non-renewable energy [4].

Biomass is an important energy source in renewable energy because it can achieve carbon neutrality. Biomass has many advantages, such as diversity, practicality and sustainability [5]. Biomass can be turned into usable energy through a variety of methods, including physical and chemical methods. Examples of biomass include plants, forest waste, animal waste and municipal solid waste, etc. [6]. According to the World Bioenergy Association statistical report, biomass accounted for 9.5% of all energy supply and contributed about 55.6 EJ of energy in 2017. The greatest contribution to bioenergy belongs to the forestry sector, where forestry biomass has contributed up to 85% of the total energy [7,8].

Bioenergy is obtained mostly from wood biomass, and wood-based bioenergy development can play a vital role in achieving energy independence, reducing carbon emissions and promoting rural development [9]. With the advent of advanced afforestation treatments and efficient biotechnology, wood-based bioenergy development can meet the needs of sustainable energy production [10]. About 11% of the world’s primary energy consumption comes from biomass. However, material shortage in wood-based biomass urgently needs an alternative source for bioenergy production. For example, *Eucalyptus pellita* and *Hevea brasiliensis* clones are potential woods for bioenergy production, with a net calorific value of 16,502 kJ kg^−1^ and 19,757 kJ kg^−1^_,_ respectively [11]. *Tachigali vulgaris* planted at a land greater than 6 m^2^ has a net calorific value more than 7.95 MJ/kg, and a medium basic density which is suitable for bioenergy production [12]. Buss et al. found that using native willow fragments for 2–37 years can yield greenhouse gas benefits within 0–20 years in Fort McPherson, Northwest Territories, Canada [13]. *Pinus* spp. and *Quercus* spp. were pressed into solid biofuel blocks that can be used to meet the need to generate low-power heat for the residential sector. The preparation of solid biofuel blocks from biomass residues is efficient, economical and easy to manufacture and use [14].

Lignocellulosic biomass is the most widespread biological resource on earth and can be transformed into value-added by-products, including biomass materials, bio-oils and biofuels [15,16]. Lignocellulosic biomass has been well recognized in the production of chemicals and wood biomaterials [17]. Lignocellulosic biomass can be found in almost all types of plants, and it does not affect food security or biodiversity or cause pollution to the environment [18,19]. Cellulose, hemicellulose and lignin make up lignocellulosic biomass; it is a complex polymer material. [20,21,22]. Lignocellulosic biomasses are rich in phenols, carbohydrates, lipids and pectin, which has the potential to be converted into valuable liquid or gaseous biofuels [23]. Lignocellulosic materials can also be converted into bioethanol or biomethanol after treatment [24,25]. The utilization of lignocellulosic biomass can help to improve the forestry economy and generate additional income for society, as well as create employment opportunities for people [26,27].

*Staphylea holocarpa* Hemsl. (*S. holocarpa*) is an endemic plant in China; it is a small deciduous tree belonging to staphyleaceae. *S. holocarpa* has compound leaves (three leaflets), and smooth and glabrous branchlets. Its flowers are pear-shaped bulging capsules, and usually pink in color. It blooms from April to May and ripens in September, which is of ornamental value [28]. *S. holocarpa* can be found throughout China, including in Henan, Anhui and Shanxi, generally on limestone slopes of deciduous forests at altitudes of about 900–1000 m [29]. The bark of *S. holocarpa* is rich in cellulose. The seeds are pressed and used in soap and paint production [30]. *S. holocarpa* root has the functions of moisturizing the lungs and relieving coughs, dispelling wind and dampness; this is of great developing value in the utilization of wild economic plants [31]. Tian studied the diverse distribution of *S. holocarpa* plants, including 13 distribution types and 10 variations. According to the distribution area types of the genus, *S. holocarpa* has obvious temperate characteristics. With the increase in altitude, the population structure changed from growing to stable [32].

To the best of author’s knowledge, limited research has been reported about *S. holocarpa* wood, especially in the field of bioenergy. Therefore, this article begins with *S. holocarpa* wood as a subject of research on its latent as a lignocellulosic bioenergy source. The active component of *S. holocarpa* wood was extracted and was characterized with FT–IR, GC–MS and TGA. The Py/GC–MS and NMR were used to detect the pyrolysis product and characterize the composition, functional groups and thermal stability from pyrolysis of *S. holocarpa* wood.

## 2. Results and Discussion

### 2.1. FT–IR Analysis

In this experiment, the FT–IR of four *S. holocarpa* wood extracts were analyzed in the range of 4000–550 cm^–1^ (Figure 1). Different functional groups were observed in the four extracts. All samples showed bands around 3376 cm^–1^, indicating tensile vibrations of intermolecular H–O–H [33,34]. The tensile vibrations of –CH2– in alkanes were mainly distributed in the range of 2976–2837 cm^–1^ [35]. Due to the presence of ester acids and aromatic components, there were C=C and C=O stretching at 1655 cm^–1^ [36]. Bands close to 1927 cm^–1^ also represent =O stretching, and the high wave number may occur due to the induced effect of C–F substituents. The absorption peaks at 1454, 1414, 1412 and 1380 cm^–1^ belong to C-H stretching [37]. There is a small absorption peak at 1275 cm^–1^ in *S. holocarpa* wood ethanol, ethanol/benzene and ethanol/methanol extracts, but not in *S. holocarpa* wood methanol extract, representing –CH_3_ bonds [38]. Due to –O vibration, there were overlapping peaks between 1090 and 1115 cm^–1^ [39]. In particular, the deep bands at 1026 cm^–1^ (A1 extract) and 1051 cm^–1^ (the other three extracts) indicate –O stretching in the tetrahedral sheet [40]. The formation of peaks at 1090 and 1053 cm^–1^ was due to C–O–C and –OH vibrations. Thus, –OH, together with the peak assigned to the aromatic ring (1655 cm^–1^), indicates the presence of phenols [41].

All spectra exhibit similar spectral signatures, except for differences in infrared absorption intensity. Typical aromatic bands of lignin were at 1655 and 1454 cm^–1^ (Figure 1) [42]. These samples were rich in carbon, because the transmission intensity of each peak gradually increases as the type of carbon changes. In contrast, the number of absorption peaks in *S. holocarpa* wood ethanol, ethanol/benzene and ethanol/methanol extracts were greater than in *S. holocarpa* wood methanol extract, and absorption peaks were associated with –CH_3_ bonds [43]. The FT–IR spectrum showed no –C≡C– vibrations at 2140–2100 cm^–1^, indicating the absence of a C≡C functional group in *S. holocarpa* wood (Figure 1). Another possible reason is that during the extraction process, some chemical bonds become unstable, or condense under high temperature conditions [44]. In general, the main chemical components of *S. holocarpa* wood samples characterized by FT–IR testing include phenols, alcohols, acids, and hydrocarbons. The absorption peaks of the four sample extracts were mainly distributed at 3800–3030 cm^–1^, 3030–2835 cm^–1^, and 1500–881 cm^–1^.

### 2.2. GC–MS Analysis

GC–MS detection results show that 15, 37, 27 and 44 chemical components were identified from four extracts of *S. holocarpa* wood, which were ethanol, methanol, benzene/ethanol, and ethanol/methanol, respectively (Figure 2). More specifically, they were as follows: 2-furanmethanol (1.37%), dihydroxyacetone (8.88%), 5-hydroxymethylfurfural (20.26%), 1,2,3-propanetriol, 1-acetate (6.55%), D-galactose (2.84%) (Appendix A). Furfural (4.99%), butyl 2-acetoxyacetate (5.47%), 4-nonanol (2.36%), sucrose (4.91%), dl-.alpha.-tocopherol (20.75%) (Appendix A). 1-hexanol, 2-ethyl- (47.63%), butyl 2-acetoxyacetate (4.01%), dibutyl phthalate (4.72%), 4-((1E)-3-hydroxy-1-propenyl)-2-methoxyphenol (3.94%), 2-propenoic acid, 3-(4-hydroxy-3-methoxyphenyl)- (0.56%) (Appendix A). 2-furanmethanol (1.48%), ethanone, 1-(2-hydroxy-5-methylphenyl)- (3.04%), n-hexadecanoic acid (2.23%), linoelaidic acid (3.86%), 1,4-bis(trimethylsilyl)benzene (4.68%) (Appendix A).

It can also be seen from Appendix A that the identified compounds can be divided into acids, alcohols, ketones, etc. *S. holocarpa* wood mainly contains 3-(4-hydroxy-3-methoxyphenyl)-2-acrylic acid, which has the potential to inhibit thrombosis, reduce inflammation, inhibit tumors and enhance sperm vitality. Clinically, 3-(4-hydroxy-3-methoxy-methoxy)-2-acrylic acid is mainly used for the vaso-assisted treatment of atherosclerosis, coronary heart disease, cerebrovascular disease, glomerular disease, pulmonary hypertension, etc., for phenyl. Furthermore, 3-(4-hydroxy-3-methoxy-methoxyacid)-2-acrylic acid can be used to treat migraines and vascular headaches, enhance hematopoietic function and treat leukocytopenia and thrombocytopenia [45,46,47]. dl-.alpha.-Tocopherol has many beneficial effects, such as eliminating pigment deposits in cells, slowing cell aging and promoting protein renewal synthesis, wound healing and the proliferation of capillaries and small blood vessels. In addition, it can improve the surrounding blood circulation and increase the oxygen supply in the tissues, thus creating good nutritional conditions for the healing of ulcers and antioxidants to protect cells from oxidative stress or damage [48,49].

In addition, many chemical components that are conducive to the development of biomass energy were detected in the four extracts. For example, furfural is an important biomass-derived platform molecule that can be used to synthesize a variety of value-added chemicals. Furfural and its derivatives are promising alternatives to traditional petrochemicals [50]. The furfural industry is constantly evolving. Recently, the annual global production of furfural exceeded 300,000 tons, of which about 70% was produced in China [51]. Furfural and its derivatives are widely used in industrial production in organic solvents, pharmaceuticals, agricultural chemicals, biofuels and fuel additives [52]. One of the most important value-added products obtained from glycerin is dihydroxyacetone. Dihydroxyacetone can also be used as a building block in organic synthesis and is a promising area for the development of novel polymer biomaterials. One example is the design of injectable synthetic biodegradable polymer biomaterials composed of polyethylene glycol and a polycarbonate of dihydroxyacetone [53,54]. Among the various biomass-derived chemicals, 5-hydroxymethylfurfural has received great attention due to its potential applications, and is listed by the U.S. Department of energy as a promising platform chemical [55]. 5-hydroxymethylfurfural is a high-value central platform chemical that can be obtained directly from hexose dehydration. The unique structure of 5-hydroxymethylfurfural gives it high chemical activity and allows it to be transformed through various catalytic processes such as oxidation, hydrogenation and amination. It can be used in the production of high value-added chemicals and liquid fuels such as 2,5-furandialdehyde, 2,5-furandicarboxylic acid, levulinic acid, etc. [56]. In general, the chemical components identified via GC–MS from the *S. Holocarpa* wood extracts shows its potential in biomedicine and bioenergy. Thus, *S. Holocarpa* wood has the potential to be used as a lignocellulosic biomass source for bioenergy production.

### 2.3. TGA Analysis

The decomposition process of *S. holocarpa* wood was studied by TGA method in the range of 30–300 °C (rate of 20 °C/min). The changes in sample mass (TGA) and thermal degradation rate (DTG) are shown in Figure 3. According to the TGA curve, there were two distinct heat loss phases in *S. holocarpa* wood. The first stage occurs at temperatures around 83 °C and the weight of the wood was slightly reduced by 3.33%. As shown in the DTG curve, the maximum mass loss rate for this stage occurs at the first peak of 54 °C. There were reports that this stage was a drying process, where mass loss represents the removal of moisture and volatiles [57]. The weight reduction of *S. holocarpa* wood (3.33%) indicates that the wood has a certain moisture. The second phase of the mass reduction occurred in the range of 190–300 °C, and the wood mass was reduced by a total of 20.63% [58].

The decomposition temperature of the wood biomass was about 190 °C; however, this decomposition temperature was delayed compared to some reported plants, such as *J. nudiflorum* wood biomass [59]. The DTG curve has a peak around 300 °C, indicating that weight loss was fastest at this temperature. This stage was the active pyrolysis zone which belongs to the main stage of the volatile stage and pyrolysis mass loss [60]. The mass reduction was mainly due to the breakdown of lignocellulose’s organic components. At this stage, the mass changes significantly, possibly caused by the changes in the chemical structure; chemical composition macromolecules were rapidly decomposed into more volatile small molecules at high temperatures [61]. The mass loss in the whole process of 0–300 °C is only 23.96%, and the heat loss is small, indicating that *S. holocarpa* wood has good thermal stability. In addition, the temperature set by this project was far from the carbonization temperature > 300 °C. Therefore, more volatile components can be obtained using this process [62].

### 2.4. Py/GC–MS Analysis

A total of 214 compounds were identified based on Py/GC–MS results (Figure 4). Among the compounds, the products of *S. holocarpa* wood pyrolysis at 500 °C were: ethyne, fluoro- (5.96%), dihydroxyacetone (4.22%), acetaldehyde (3.31%), ethyl ether (2.81%), hexadecanenitrile (2.33%), 2-propanone, 1-hydroxy- (2.11%), methyl glyoxal (2.07%), furfuryl alcohol, tetrahydro-5-methyl-, cis- (2.05%), etc. (see Appendix A).

According to the statistics of Appendix A, there are seven categories of ketones (48, 20.35%), aldehydes (13, 9.09%), acids (10, 3.09%), esters (18, 6.42%), alcohols (40, 15.64%), phenols (18, 18.69%), and ethers (5, 0.79%), of which the proportion of ketones (10.31%), alcohols (13.66%) and phenols (27.89%) were higher (Figure 5a). The pyrolysis of *S. holocarpa* wood is divided into three stages according to the time: <5 min, 5–25 min and >25 min; the pyrolysis products account for 10.795%, 47.546% and 41.695%, respectively (Figure 5b). In the <5 min stage, most of the pyrolysis products were small molecules of organic acids. In the 5–25 min stage, the pyrolysis products were mainly ketone compounds, and the reaction types were mainly double-bond reductions due to the presence of C=C and C=O. Most furan and cyclopentenones come from hemicellulose [63]. In the stage >25 min, the major component was phenolic substances and their derivatives produced by lignin pyrolysis. At the stage of 5–25 min, the pyrolysis products of the sample were the highest, indicating that the ketone content was high [64]. According to the properties of compounds at different stages, most of the compounds identified were organic acids, ketones, furans, cyclopentenes, phenols and their derivatives [65]. Most of these compounds are used in biopharmaceutical, chemical and energy industries [66,67].

Many of the pyrolysis products identified by Py/GC–MS detection can be used in the chemical industry as green energy. For example, acetaldehyde belongs to biomass-derived oxygenated compounds, which are one of the main components of bio-oil. Acetaldehyde is mainly used as a reducing agent and is industrially used in the manufacturing of polyacetaldehyde, acetic acid, synthetic rubber, etc. [68]. Formic acid is a major product of carbohydrates derived from biomass and is receiving increasing attention as a sustainable hydrogen source. Formic acid-mediated biomass feedstock can be converted into value-added products, including biofuels, levulinic acid, etc. [69]. Catechol is an industrially relevant chemical with countless applications. It is the most representative basic structure unit in lignin, and it is also the main reaction intermediate and product in biomass or lignin pyrolysis [70]. Catechol plays an important role in many systems by interacting with organic and inorganic compounds. In addition, catechol crosslinked polymer networks exhibit remarkable mechanical strength, good adhesion and realistic properties [71].

Biomass provides an important source of raw materials, and is ideal for the development of functional or intermediate molecules for chemical synthesis, such as glycerol carbonate or glycidol [72]. Maltol is one of the derivatives of biomass, and maltol by-products have a certain synergistic effect with pine chips. Adding less than 10% maltol by-products to pine wood chips to make a fuel blend can improve combustion characteristics and reduce emissions [73]. 1,2-cyclopentanedione, 3-methyl- is an orthocyclodione, which is an important fine chemical intermediate and is widely used in pharmaceutical, chemical and other industries [74]. Phenol, 2-methyl- can be used in organic synthesis and also as a disinfectant and preservative; it is an important pharmaceutical intermediate. It is also the main compound in bio-oils [75]. Creosol is a lignin derivative of biomass and is a high value-added product, as a source of renewable assets of great interest to industry [76,77]. Similarly, the thermal cracking products detected by Py/GC–MS also contain chemical components such as furfural [50,51,52], dihydroxyacetone [53,54], and 5-hydroxymethylfurfural [55,56]. Analysis of pyrolysis products shows that the chemical components in *S. holocarpa* wood can be well applied in chemical, bioenergy and other fields. At the same time, the test results of Py/GC–MS and GC–MS were also consistent, which further demonstrates the potential of *S. holocarpa* wood for use as a source for bioenergy production.

### 2.5. NMR Analysis

#### 2.5.1. ^1^H–NMR Analysis

The ^1^H–NMR spectrum of the nuclear magnetic resonance spectrum is the most widely used. It not only offers high magnetic detection sensitivity, but the signal is easy to observe; organic compounds provide a large number of hydrogen atoms in a variety of chemical environments. As can be seen from Figure 6, the overall displacement ranges from 0–8 ppm. Typically, the chemical shifts of the protons were saturated at δ 0.2 to δ 1.5 alkane compounds; usually, the first proton appears at about δ 0.9, the second proton appears at δ 1.3, and the third proton appears at δ 1.5 [78]. Generally, the proton chemical shift of the carbon atom directly connected to the halogen between 2.0 and 4.5, and the influence of the proton on the adjacent carbon atom was significantly decreased. The chemical shift of α–C protons near carbonyl or cyano was 2–3 ppm, which was caused by the anisotropic effect of C=C and sp^2^ hybridization of internal changes in olefin carbon. The chemical shift in the range of 4.5–5.9 ppm belongs to olefin compounds, and the δ value increases after coupling with the aryl group [79]. Chemical shifts of 1.7–3.5 ppm were alkyne carbon protons. The chemical shifts of the carbonyl α-H belong to carbonyl compounds between 2 and 2.7 ppm, and the chemical shifts of the acid compounds range between 2 and 2.6 ppm. Displacement in the 3.3–4 ppm range is mainly caused by α-H hydrogen atoms in the ether molecule. The chemical shift of the α-H hydrogen atom in the alcohol molecule was 3.4–4 ppm, and the displacement of ester compounds and phenolic hydroxyl protons was 2–4.1 ppm and 4–8 ppm, respectively. The aromatic compound chemical shifts range from 6.3 to 8.5 ppm, and the heterocyclic aromatic protons range from 6.0 to 9.0 ppm [80].

The results of ^1^H–NMR detection showed that *S. holocarpa* wood was rich in chemical components, including acids, ethers, alcohols, esters, aromatics and other organic compounds. Bio-oil is a complex mixture of highly oxygenated organic components, including almost all types of oxygenated organic compounds. *S. holocarpa* wood has the potential to become a green and sustainable energy source. This was consistent with FT–IR, GC–MS, TGA and Py/GC–MS test results.

#### 2.5.2. ^13^C–NMR Analysis

Organic elements are the skeleton of carbon, currently ^13^C–NMR spectroscopy is used to study the structure of sample changes through carbon atoms to understand the structure of organic compounds. Since some of the functional groups in the organic compound do not contain hydrogen atoms, these functional groups cannot be obtained from the ^1^H spectrum and can only be obtained from the ^13^C spectrum (Figure 7). For carbohydrates, the polymer carbon is mainly distributed in the 50–110 ppm region [81]. The carbon spectrum of saturated hydrocarbons, partial alkynes and partial olefins were mainly distributed between 15–45 ppm and 75–55 ppm. The carbon spectrum containing halogen elements (C-I: 0–40 ppm; c-bromine: 25–65 ppm; C-ci: 35–80 ppm) and the carbon spectrum of aromatic and olefins (=C–: 100–150 ppm; C_6_H_6_: 110–160 ppm) were mainly distributed between 0–80 ppm and 10–160 ppm. The signals at 20.3 ppm and 172.9 ppm shifts were mainly assigned to hemicellulose acetyl [82,83,84].

#### 2.5.3. 2D–HSQC Analysis

In order to further understand the sample structure in more detail, 2D–HSQC NMR analysis was also carried out in this study. 2D–HSQC NMR is currently one of the most widely used technologies. Its ability to categorize functional groups clearly from 2D–HSQC maps provides an important structural shift in the sample and provides a resolution of overlapping signals in both ^1^H and ^13^C–NMR spectra, and analysis of two different aromatic groups [85]. In Figure 8, the aliphatic region of the HSQC spectrum was located at δC/δH10–40/0.5–2.5, the side chain was at δC/δH50–95/2.5–6.0 and the aromatic was at δC/δH95–135/5.5–8.0 [86]. δC/δH92/5.3, δC/δH96/4.4δC/δH102/6.6, δC/δH110/6.9, δC/δH115/6.7, and δC/δH120/6.8. These chemical shifts indicate the interconnection between hydrogen atoms and carbon atoms [87]. As can be seen from the 2D–HSQC NMR chart, the linking of saturated hydrocarbons was at δC/δH15–45/0.2–1.5, the carbon spectra of partial alkynes and alkenes were δC/δH55–75/1.7–3.5, 4.5–5.9, and the aromatic and olefinic carbons are located at δC/δH110–140/6.3–8.5, 4.5–5.9, while the remaining alcohols, ethers, phenols and other oxygenates are located at δC/δH 60–80/3.3–4 [88,89].

## 3. Material and Methods

### 3.1. Sources of Sample and Extractives

*S. holocarpa* wood was collected from Xiaoqinling National Forest Park, Henan Province, and the wood sample was obtained from the trunk. The sample was processed into two different particle sizes, which were 40–60 μm and 200 μm powder. The powder with particle size of 40–60 μm was prepared for extraction, while the 200 μm powder was prepared for pyrolysis. About 15 g of the 40–60 µm powder was weighted, and extraction was performed in different organic solvents. Four types of organic solvent were prepared with optimum extraction temperature, including ethanol (75 °C), methanol (65 °C), benzene/ethanol (1:1) (90 °C) and ethanol/methanol (1:1) (70 °C). The extraction was performed for 5 h. After extraction, the four extracts were separated, and the excess solvent was removed with a rotary evaporator. The concentrated extracts were now ready for FT–IR and GC–MS analysis.

### 3.2. Experiment Methods

#### 3.2.1. FT–IR Analysis

About 1 mL of extractives and 1 g KBr were mixed and pressed into KBr tablets which were, respectively, detected using a Thermo Fisher Nicolet iS10 FT–IR spectrophotometer (Waltham, MA, USA) by KBr method [90].

#### 3.2.2. GC–MS Analysis

Each of the four extracts was tested on the Agilent–7890B–5977A (Santa Clara, CA, USA) instrument. Elastic quartz capillary columns (HP–5MS: 30 m × 250 μm × 0.25 μm) were used. The carrier gas flow rate was 1 mL/min; the carrier gas used was high purity helium, and the split ratio was set at 2:1. First, the temperature rises gradually from 50 °C to 250 °C at a rate of 8 °C/min, and then from 250 °C to 300 °C at a rate of 5 °C/min. The program was set to scan quality ranges, ionization voltages, ionization currents, ion sources and four-pole data for 30–600 amu, 70 eV, 150 ua, 230 °C and 150 °C [91].

#### 3.2.3. TGA Analysis

Analysis of the 1 mg 200 μm powder was performed on a Perkin Elmer–STA8000 (Waltham, MA, USA) instrument. The temperature was set to rise from 30 °C–300 °C at the rate of 5 °C/min. The carrier gas rate was set at 60 mL/min and the gas used was nitrogen [92].

#### 3.2.4. Py/GC–MS Analysis

About 0.1 mg of the powder sample (200 mu) was placed in an Agilent–7890B–5977A CDS5000 (Santa Clara, CA, USA) instrument for analysis and detection. The carrier gas, pyrolysis temperature, heating rate and pyrolysis time used were helium (high purity), 500 °C, 20 °C/ms and 15 s, respectively. The capillary column (HP-5MS: 60 m × 250 μm × 0.25 μm), pyrolysis product delivery line, injection valve temperature, shunt ratio and shunt velocity were 300 °C, 300 °C, 1:60 and 50 mL/min, respectively. The GC was programmed to last for 2 min at 40 °C, increasing from 40 °C to 120 °C at a rate of 5 °C/min. At 10 °C/min, it went from 120 °C to 200 °C for 15 min. The MS ion source temperature and scanning range was 230 °C and 28–500 amu. [93].

#### 3.2.5. NMR Analysis

The NMR polarizer model is Agilent–400 MR (Santa Clara, CA, USA) and methanol-d4 was the solvent used in the testing process. The entire assay was performed with the same NMR probe: ^1^H–NMR, ^13^C–NMR, and 2D–NMR. The duration, sample-and-hold time, pulse, pulse width and frequency of ^1^H–NMR were 1.000 s, 2.556 s, 45 degrees, 6410.3 Hz and 399.79 MHz, respectively. The duration, sample-and-hold time, pulse, pulse width, frequency, hydrogen decoupling frequency and power of ^13^C–NMR are 1.000 s, 1.311 s, 45 degrees, 25,000 Hz, 100.53 MHz, 399.79 MHz and 38 dB, respectively. The duration, sample-and-hold time, pulse, pulse width, frequency, frequency of carbon decoupling and power of 2D–HSQC are 1.000 s, 0.150 s, 4807.7 Hz, 20,105.6 Hz, 399.79 MHz, 100.54 MHz and 38 dB, respectively [94].

## 4. Conclusions

This project utilized *S. holocarpa* wood as the research object to evaluate its potential as a lignocellulosic biomass source for bioenergy production The results proved that *S. holocarpa* wood contains a large amount of organic chemicals that can be used in the bioenergy and chemical industries. This project also provides the research basis of *S. holocarpa* wood pyrolysis, in which a large number of volatile compounds (including ketones, alcohols and phenol) were detected, which are proven to be important organic components of bio-oil. The NMR detection method used to identify the distribution of functional groups also proved that *S. holocarpa* wood has the potential to become a biomass energy source. This study explores the relationship between potential of the extraction and the pyrolysis of *S. holocarpa* wood, which is helpful for the exploitation and utilization of *S. holocarpa* wood as for bioenergy production. For the first time, NMR technology has been applied to *S. holocarpa* wood, and pyrolysis products show the potential to become high-value products. In the future, pyrolysis with various parameters can be optimized to maximize bioenergy production.

## Figures and Tables

**Figure 1 molecules-28-00299-f001:**
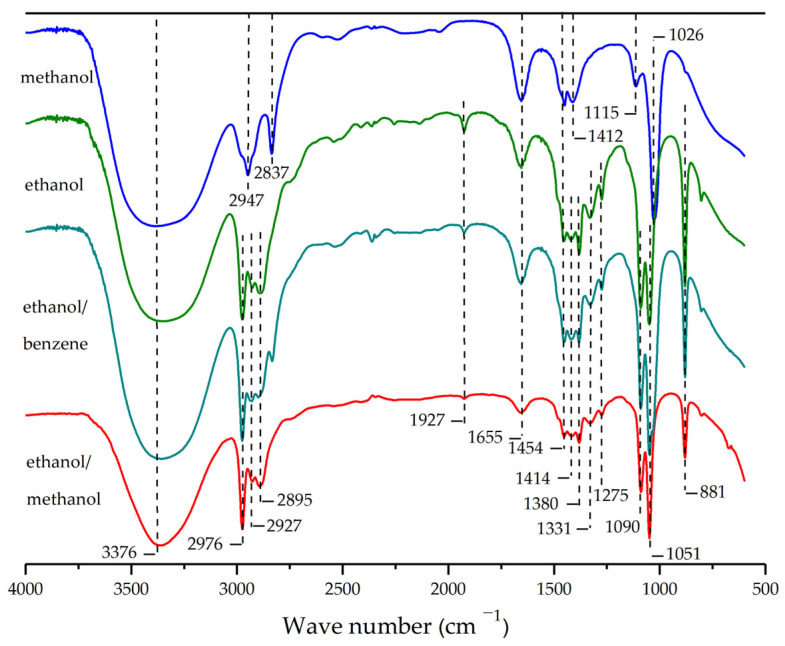
FT–IR spectra of four extract samples of *S. holocarpa* wood. Methanol, ethanol, ethanol/benzene, and ethanol/methanol extract samples are represented by the blue line, green line, blue-green line, and red line, respectively.

**Figure 2 molecules-28-00299-f002:**
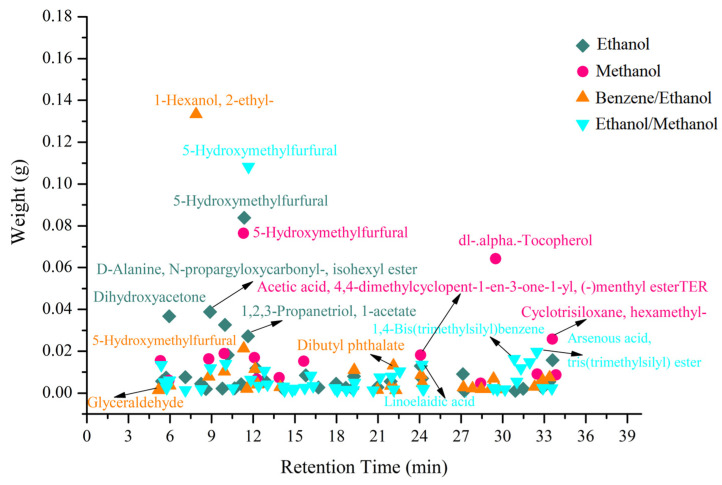
Total ion chromatograms of *S. holocarpa* wood four extract samples. Ethanol, methanol, benzene/ethanol, and ethanol/methanol extract samples are represented by the green square, purple circle, yellow triangle and blue triangle, respectively.

**Figure 3 molecules-28-00299-f003:**
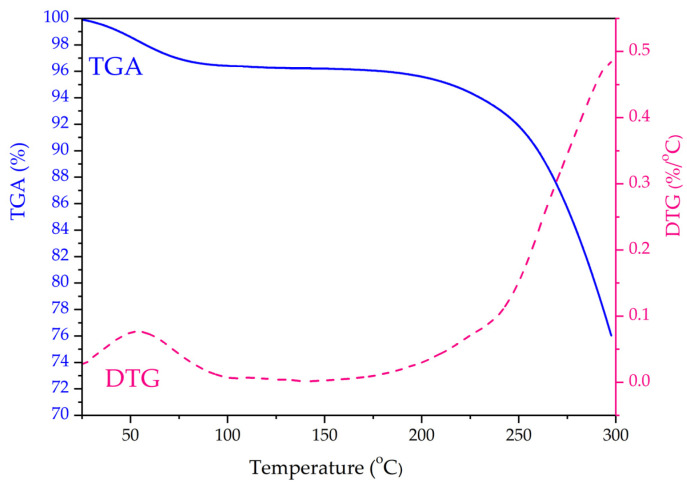
TGA–DTA curve of *S. holocarpa* wood.

**Figure 4 molecules-28-00299-f004:**
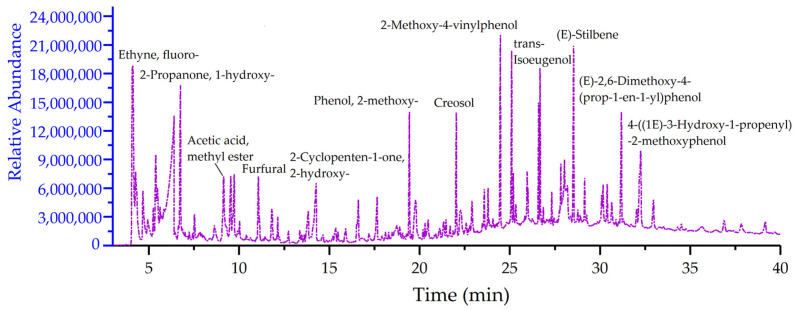
The total ion chromatography of *S. holocarpa* wood was determined by Py/GC–MS.

**Figure 5 molecules-28-00299-f005:**
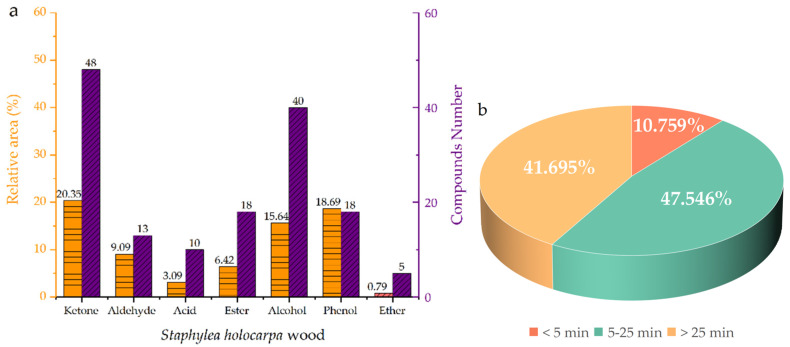
Classification (**a**) and temporal (**b**) distribution of pyrolysis products from *S. holocarpa* wood.

**Figure 6 molecules-28-00299-f006:**
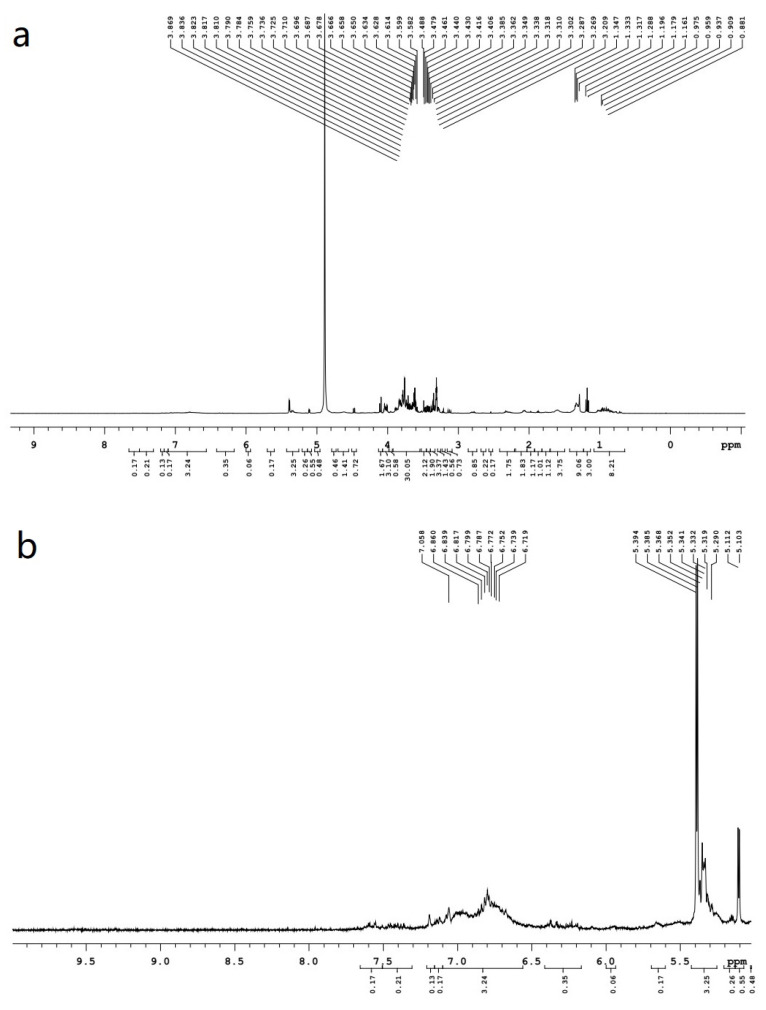
Total distribution of ^1^H–NMR of *S. holocarpa* wood; (**a**–**d**) are enlarged views.

**Figure 7 molecules-28-00299-f007:**
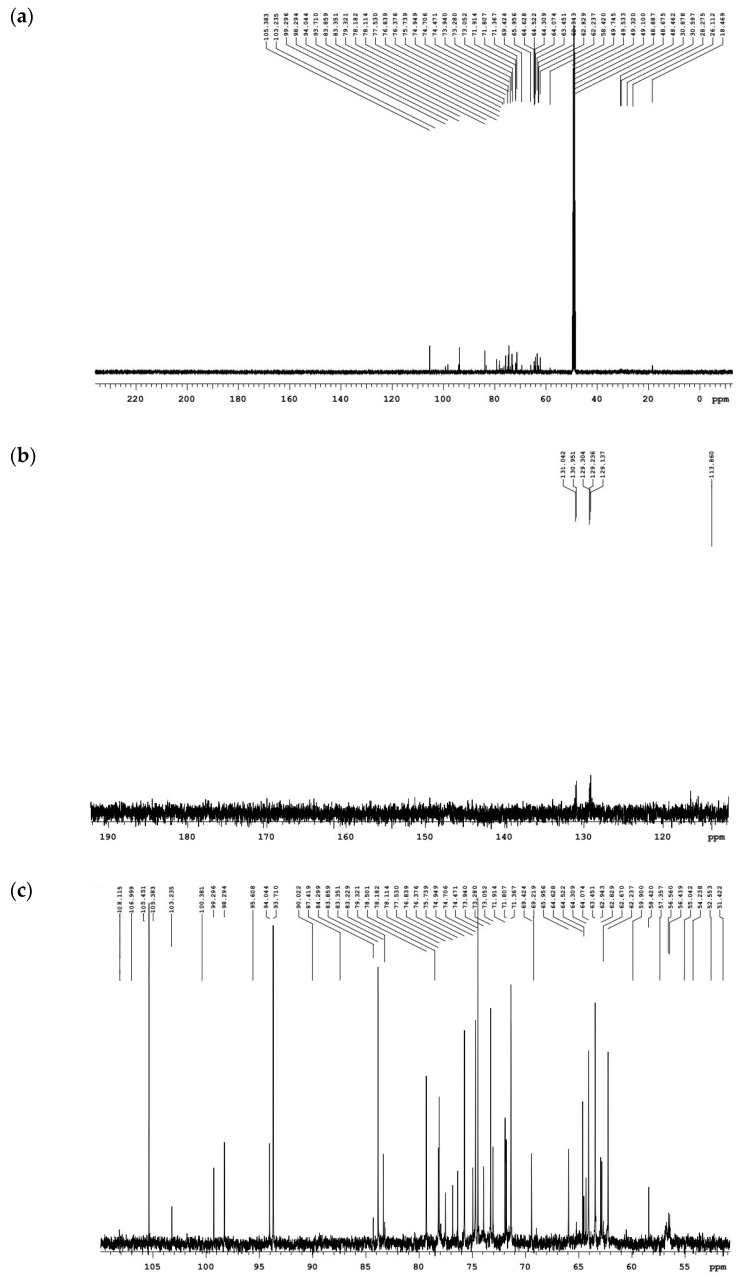
^13^C–NMR total distribution of *S. holocarpa* wood; (**a**–**d**) show enlarged views.

**Figure 8 molecules-28-00299-f008:**
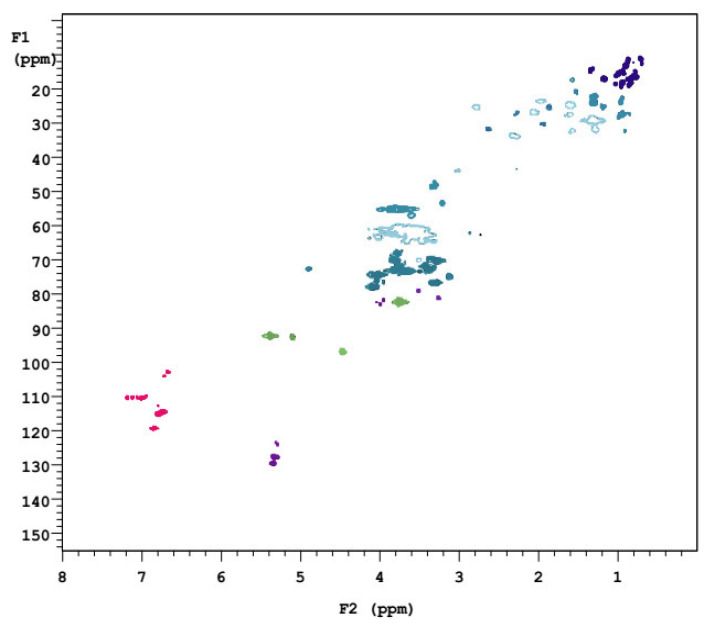
2D–HSQC NMR spectra of the *S. holocarpa* wood sample.

## Data Availability

The data presented in this study are available on request from the corresponding author.

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
