# Peer review of "Potential of *Staphylea holocarpa* Wood for Renewable Bioenergy"

_molecules, 2022, doi:10.3390/molecules28010299_

Round 1
Reviewer 1 Report
In this study, wood samples of S. holocarpa were analyzed using various instruments to determine the chemical components of various wood extracts and combustion/pyrolysis products. The results will be useful. Following are some of the review comments on this manuscript.
In the manuscript title, the word “Explore” does not fit well; this can be changed to “Exploring”. Alternatively, the terms “Explore the” may be removed from the title, which will not change the intended meaning of the title.
Section 2.1: In this section, the age of the plant (tree), and whether the wood sample was taken from the main trunk wood or from the branches or twigs, etc. should be provided.
Section 3.5.213. C- NMR analysis: This section just repeats the information from the previous Section 3.5.11. H-NMR analysis. This is a major error. Thus, the lines 262-281 should be removed, and the results of Figure 7 should be discussed here. Again, this is a major comment in this manuscript.
There are several English language and punctuation errors throughout the manuscript. Below are few example errors. All of the English errors should be corrected in the revision with a help of a professional English language editor.
Line 55: “Cellulose hemicellulose and lignin”.
Lines 75-76: This sentence is not grammatically correct: “Due to the impact on S. holocarpa wood's research is less and largely unreported, especially in bioenergy.”
Lines 85-86: The procedures should be written in Passive Voice. The following sentence is not acceptable: “Weigh 4 parts of 40-60 μm powder, 15g each, and extract them in 300 ml of organic solvent.”
Line 242: “environments are provide”.
Author Response
Manuscript Number: molecules-2099583
Title: Potential of Staphylea holocarpa wood for renewable bioenergy
Dear respected editor and reviewers, we would like to express our sincere gratitude to you for writing us the following constructive comments on our manuscript. Also, we appreciate very much for your willingness to check and help to improve the overall contents and quality of our manuscript with your precious time. Thank you so much for your comments and advice. We have made our best efforts to revise and improve our manuscript in an effort to acknowledge the reviewers’ comments accordingly. The comments from the reviewers are retyped below in italics, our responses are typed in normal black font, and the modifications done to the manuscript are also shown in red font. Thank you very much.
Comments from Reviewers
Reviewer #1:
In this study, wood samples of S. holocarpa were analyzed using various instruments to determine the chemical components of various wood extracts and combustion/pyrolysis products. The results will be useful. Following are some of the review comments on this manuscript.
Answer: We thank you for your careful review and for given us a possibility to improve the quality of our manuscript.
Q1. In the manuscript title, the word “Explore” does not fit well; this can be changed to “Exploring”. Alternatively, the terms “Explore the” may be removed from the title, which will not change the intended meaning of the title.
Answer: We thank the reviewer very much for the comments. We changed the title in the manuscript.
Action: “ Potential of Staphylea holocarpa wood for renewable bioenergy.”
Q2. Section 2.1: In this section, the age of the plant (tree), and whether the wood sample was taken from the main trunk wood or from the branches or twigs, etc. should be provided.
Answer: We thank the reviewer very much for the comments. Because we are studying this species, there is no restriction on the age of the tree. The wood sample was taken from the main trunk wood.
Action: “ 2.1. Sources of Sample and Extractives.
- holocarpa wood was collected from Xiaoqinling National Forest Park, Henan Province,and the wood sample was taken from the main trunk wood. The sample was processed into two different sizes of 40-60 u and 200 u powder......”
Q3. Section 3.5.213. C- NMR analysis: This section just repeats the information from the previous.
Answer: We thank the reviewer very much for the comments. We corrected Section 3.5.2. 13C- NMR analysis.
Action: “ 3.5.2. 13C- NMR analysis
Organic elements are the skeleton of carbon currently 13C–NMR spectroscopy is used to study the structure of sample changes through the carbon atoms to understand the structure of organic compounds. Since some of the functional groups in the organic compound do not contain hydrogen atoms, these functional groups cannot be obtained from the 1H spectrum and can only be obtained from the 13C spectrum (Figure 7). For carbohydrates, the polymer carbon is mainly distributed in the 50–110 ppm region [85]. The carbon spectrum of saturated hydrocarbons, partial alkynes and partial olefins were mainly distributed between 15–45ppm and 75–55ppm. Carbon spectrum containing halogen elements (C-I: 0–40ppm; c-bromine: 25–65 ppm; C-ci: 35–80 ppm), carbon spectrum of aromatic and olefins (= C–: 100–150 ppm; C6H6: 110–160 ppm) were mainly distributed in 0–80 ppm and 10–160 ppm. And the signals at 20.3 and 172.9 ppm shifts are mainly assigned to hemicellulose acetyl [86–88].”
Q4. Section 3.5.11. H-NMR analysis. This is a major error. Thus, the lines 262-281 should be removed, and the results of Figure 7 should be discussed here. Again, this is a major comment in this manuscript.
Answer: We thank the reviewer very much for the additional proposal. We have revised this section in the manuscript.
Action: “ 3.5.2. 13C- NMR analysis
Organic elements are the skeleton of carbon currently 13C–NMR spectroscopy is used to study the structure of sample changes through the carbon atoms to understand the structure of organic compounds. Since some of the functional groups in the organic compound do not contain hydrogen atoms, these functional groups cannot be obtained from the 1H spectrum and can only be obtained from the 13C spectrum (Figure 7). For carbohydrates, the polymer carbon is mainly distributed in the 50–110 ppm region [85]. The carbon spectrum of saturated hydrocarbons, partial alkynes and partial olefins were mainly distributed between 15–45ppm and 75–55ppm. Carbon spectrum containing halogen elements (C-I: 0–40ppm; c-bromine: 25–65 ppm; C-ci: 35–80 ppm), carbon spectrum of aromatic and olefins (= C–: 100–150 ppm; C6H6: 110–160 ppm) were mainly distributed in 0–80 ppm and 10–160 ppm. And the signals at 20.3 and 172.9 ppm shifts are mainly assigned to hemicellulose acetyl [86–88].”
There are several English language and punctuation errors throughout the manuscript. Below are few example errors. All of the English errors should be corrected in the revision with a help of a professional English language editor.
Answer: We thank the reviewer very much for the comments. Professional English language editors have revised our manuscript.
Q1. Line 55: “Cellulose hemicellulose and lignin”.
Answer: We thank the reviewer very much for the supplement. We have made modifications.
Action: “......Cellulose hemicellulose and lignin make up lignocellulose and......”
Q2. Lines 75-76: This sentence is not grammatically correct: “Due to the impact on S. holocarpa wood's research is less and largely unreported, especially in bioenergy.”
Answer: We thank the reviewer very much for the supplement. We have revised this sentence.
Action: “As little has been reported about S. holocarpa wood, especially in the field of bioenergy.”
Q3. Lines 85-86: The procedures should be written in Passive Voice. The following sentence is not acceptable: “Weigh 4 parts of 40-60 μm powder, 15g each, and extract them in 300 ml of organic solvent.”
Answer: We thank the reviewer very much for the supplement. We have revised this sentence in the manuscript.
Action: “The 40–60 mu powder was weighed 15g each, and extracted in 300 ml organic solvent, respectively. ”
Q4. Line 242: “environments are provide”.
Answer: We thank the reviewer very much for the supplement. We have revised this sentence in the manuscript.
Action: “ ......organic compounds provide a large number of hydrogen atoms in a variety of chemical environments.”

Reviewer 2 Report
Dear authors
The manuscript describes to the topic as very interesting, but I have some suggestions for improving the manuscript.
In the manuscript check, the scientific name of the S. holocarpa should be italics.
In general, the figure caption I considered should be more descriptive and more details and add the number of replicates and statistical analyses that were performed for validated the results. In addition, should be mentioned the control used in the experiments.
And in the results and discussion sections, please improve the discussion results focused on highlighting the meaning of the results about green energy renewable.
In addition, I mentioned some corrections:
methods section please used the subcategory number example 2.2.2. for the methodologies.
figure 1 please in the figure caption described and related the color with sampled analyzed.
In the experiment FT-IR what is the control positive and negative controls, it is important to mention in the figure caption and the text add in the manuscript and it is necessary to add the replicated performed experiments and statistical analyses.
Line 151, 157. please add the final sentences (Figure 1).
line 165. please correct fig.2 by Figure 2.
Line 175. please S. holocarpa should be in italics.
Line 176-190. here mentioned a discussion about the clinical benefits of metabolites found, but I considered that the paper is focused on renewable bioenergy, please I suggest adding a discussion of results in this sense.
In figure 2. Please add the description of the color and signal used for highlighting and identifying the metabolites found and add the if was performed replicates and statistical analyses.
Line 194. Please correct fig.3 by (Figure 3).
in the figure 3 caption add a description of color and data observed, it was performed by triplicates o more replicates, and mentioned that statistical analyses were performed.
Line 191-202. Please, In the section TGA analyses, please discuss the results, I did not find it.
Line 207,212. Please S. holocarpa should be in italics.
Line 220, 223. Please correct fig.5A by (Figure 5A).
Line 234-237. Please discuss more broadly what components can be used in bioenergy.
In Figure 6, please improved the figure scale, because the numbers are small.
Lines238-259. Please add the discussion results obtained.
Good luck.
Author Response
Manuscript Number: molecules-2099583
Title: Potential of Staphylea holocarpa wood for renewable bioenergy
Dear respected editor and reviewers, we would like to express our sincere gratitude to you for writing us the following constructive comments on our manuscript. Also, we appreciate very much for your willingness to check and help to improve the overall contents and quality of our manuscript with your precious time. Thank you so much for your comments and advice. We have made our best efforts to revise and improve our manuscript in an effort to acknowledge the reviewers’ comments accordingly. The comments from the reviewers are retyped below in italics, our responses are typed in normal black font, and the modifications done to the manuscript are also shown in red font. Thank you very much.
Comments from Reviewers
Reviewer #2:
Dear authors
The manuscript describes to the topic as very interesting, but I have some suggestions for improving the manuscript.
Q1. In the manuscript check, the scientific name of the S. holocarpa should be italics.
Answer: We thank the reviewer very much for the supplement. We have checked and corrected throughout the manuscript.
Q2. In general, the figure caption I considered should be more descriptive and more details and add the number of replicates and statistical analyses that were performed for validated the results. In addition, should be mentioned the control used in the experiments. And in the results and discussion sections, please improve the discussion results focused on highlighting the meaning of the results about green energy renewable.
Answer: We thank you for your careful review and for given us a possibility to improve the quality of our manuscript. We have tried our best to revise and perfect our manuscript.
Q3. In addition, I mentioned some corrections:
methods section please used the subcategory number example 2.2.2. for the methodologies.
Answer: We thank the reviewer very much for the supplement. We have modified this method section number in the manuscript.
Action: “2.2. Experiment Methods
2.2.1. FT–IR Analysis
2.2.2. GC–MS Analysis
2.2.3. TGA Analysis
2.2.4. Py/GC–MS Analysis
2.2.5. NMR Analysis”
Q4. figure 1 please in the figure caption described and related the color with sampled analyzed.
Answer: We thank the reviewer very much for the supplement. We describe the extract sample names for different colors in Figure 1.
Action: “Figure 1. FT-IR spectra of four extract samples of S. holocarpa wood. Methanol, ethanol, ethanol/benzene, and ethanol/methanol extract samples were blue line, green line, blue-green line, and red line, respectively.”
Q5. In the experiment FT-IR what is the control positive and negative controls, it is important to mention in the figure caption and the text add in the manuscript and it is necessary to add the replicated performed experiments and statistical analyses.
Answer: We thank the reviewer very much for the comments. We designed FT-IR experiments that did not involve negative or positive controls. Because the FT-IR sample we used was a liquid sample extracted from an organic solvent, only for the characterization of functional groups, we did not perform multiple replicates. As future experiments are analyzed in depth, we will take your suggestions and adjust FT-IR experiments.
Q6. Line 151, 157. please add the final sentences (Figure 1).
Answer: We thank the reviewer very much for the comments. We added the final sentence (Figure 1)
Action: “ Typical aromatic bands of lignin are at 1655 and 1454 cm–1 (Figure 1) [41].......The FT–IR spectrum showed no –C≡C– vibrations at 2140–2100 cm–1, indicating that there are no C≡C bonds in S. holocarpa wood (Figure 1)......”
Q7. line 165. please correct fig.2 by Figure 2.
Answer: We thank the reviewer very much for this question. We have changed fig.2 to Figure 2.
Action: “ GC–MS detection results show that 15, 37, 27 and 44 chemical components were identified from four extracts of S. holocarpa wood (Figure 2). ”
Q8. Line 175. please S. holocarpa should be in italics.
Answer: We thank the reviewer very much for the comments. We have changed S. holocarpa to italics.
Action: “ Figure 2. Total ion chromatograms of S. holocarpa wood four extract samples.”
Q9. Line 176-190. here mentioned a discussion about the clinical benefits of metabolites found, but I considered that the paper is focused on renewable bioenergy, please I suggest adding a discussion of results in this sense.
Answer: We thank the reviewer very much for the comments. We have added a discussion of this section.
Action: “In addition, many chemical components that were conducive to the development of biomass energy were detected in the samples of the four extracts. For example, furfural was an important biomass-derived platform molecule that can be used to synthesize a variety of value-added chemicals. Furfural and its derivatives were promising alternatives to traditional petrochemicals [55]. The furfural industry is constantly evolving. Recently, the annual global production of furfural exceeded 300,000 tons, of which about 70% was produced in China [56]. Furfural and its derivatives were widely used in industrial production in organic solvents, pharmaceuticals, agricultural chemicals, biofuels or fuel additives [57]. One of the most important value-added products obtained from glycerin was dihydroxyacetone. Dihydroxyacetone can also be used as a building block in organic synthesis and is a promising area for the development of novel polymer biomaterials. For example, the design of injectable synthetic biodegradable polymer biomaterials composed of polyethylene glycol and a polycarbonate of dihydroxyacetone [58, 59]. Among the various biomass-derived chemicals, 5-Hydroxymethylfurfural has received great attention due to its potential applications and was listed by the U.S. Department of energy as one of the promising platform chemicals [60]. 5-Hydroxymethylfurfural is a high-value central platform chemical that can be obtained directly from hexose dehydration. The unique structure of 5-Hydroxymethylfurfural gives it high chemical activity and allows it to be transformed through various catalytic processes such as oxidation, hydrogenation and amination. It can be used in the production of high value-added chemicals and liquid fuels such as 2,5-furandialdehyde, 2,5-furandicarboxylic acid, levulinic acid, etc. [61]. In general, the chemical components identified from GC–MS have very good applications in terms of biomedicine and bioenergy. Thus illustrating S. Holocarpa wood has potential as a bioenergy material.”
Q10. In figure 2. Please add the description of the color and signal used for highlighting and identifying the metabolites found and add the if was performed replicates and statistical analyses.
Answer: We thank the reviewer very much for the comments. Although we did not perform repeat experiments, the mass spectral data obtained for each component were analyzed by NIST11L standard spectral library and artificial spectra. More than 95% similarity was selected to determine the chemical composition in the standard spectrum, and the relative percentage content of each component was calculated by the peak area normalization method. We describe the extract sample names for different colors in Figure 2.
Action: “Figure 2. Total ion chromatograms of S. holocarpa wood four extract samples. Ethanol, methanol, benzene/ethanol, and ethanol/methanol extract samples were green square, purple circle, yellow triangle, blue triangle, respectively. ”
Q11. Line 194. Please correct fig.3 by (Figure 3).
Answer: We thank the reviewer very much for the comments. We have changed fig.3 to Figure 3.
Action: “ Changes in wood mass (TGA) and thermal degradation rate (DTG) are shown in Figure 3. ”
Q12. in the figure 3 caption add a description of color and data observed, it was performed by triplicates o more replicates, and mentioned that statistical analyses were performed.
Answer: We thank the reviewer very much for the supplement. Because our laboratory was being set up at the time, our TGA was tested in someone else's laboratory, so they did not repeat the test. Due to the epidemic, supplementary data is difficult to complete in a short time, can it be published or deleted according to the current data?
Q13. Line 191-202. Please, In the section TGA analyses, please discuss the results, I did not find it.
Answer: We thank the reviewer very much for the comments. We have added this section to the manuscript.
Action: “ The decomposition temperature of wood biomass was about 190 °C, but this decomposition temperature is delayed compared to some reporter plants, such as those with similar moisture content in J. nudiflorum wood biomass [64]. The DTG curve has a peak around 300 °C, indicating that weightlessness is fastest at this temperature. This stage was called the active pyrolysis zone and belongs to the main stage of volatile stage and pyrolysis mass loss [65]. The mass reduction was mainly due to the breakdown of lignocellulose's organic components. At this stage, the mass changes significantly, possibly caused by chemical changes, such as chemical composition macromolecules were rapidly decomposed into more volatile small molecules at high temperatures [66]. The mass loss in the whole process of 0–300 °C is only 23.96% and the heat loss is small, indicating that S. holocarpa wood has good thermal stability. In addition, the temperature set by this project was far from the carbonization temperature < 300 °C. So more volatile components can be obtained in this process [67]. ”
Q14. Line 207,212. Please S. holocarpa should be in italics.
Answer: We thank the reviewer very much for the comments. We have changed S. holocarpa to italics.
Action: “ Among the compounds, the products of S. holocarpa wood pyrolysis at 500 °C were......
Figure 4. The total ion chromatography of S. holocarpa wood was determined by Py/GC–MS.”
Q15. Line 220, 223. Please correct fig.5A by (Figure 5A).
Answer: We thank the reviewer very much for the guidance. We have changed fig.5a to Figure 5a.
Action: “......which the proportion of ketones (10.31%), alcohols (13.66%) and phenol (27.89%) is higher (Figure 5a). The pyrolysis of S. holocarpa wood is divided into three stages according to the time: < 5 min, 5–25 min and > 25 min, and the pyrolysis products account for 10.795%, 47.546% and 41.695%, respectively (Figure 5b). ”
Q16. Line 234-237. Please discuss more broadly what components can be used in bioenergy.
Answer: We thank the reviewer very much for the comments. We have added a discussion of this section.
Action: “Many of the pyrolysis products identified by Py/GC–MS detection can be used in the chemical industry as green energy. For example, acetaldehyde belongs to biomass-derived oxygenated compounds, which are one of the main components of bio-oil. Acetaldehyde is mainly used as a reducing agent and is industrially used in the manufacture of polyacetaldehyde, acetic acid, synthetic rubber, etc. [73]. Formic acid is a major product of carbohydrates derived from biomass and is receiving increasing attention as a sustainable hydrogen source. Formic acid mediated biomass feedstock can be converted into value-added products, including biofuels, levulinic acid, etc. [74]. Catechol is an industrially relevant chemical with countless applications. It is the most representative basic structure unit in lignin, and it is also the main reaction intermediate and product in biomass or lignin pyrolysis [75]. Catechol plays an important role in many systems by interacting with organic and inorganic compounds. In addition, catechol crosslinked polymer networks exhibit remarkable mechanical strength, good adhesion and realistic properties [76]. Biomass provides an important source of raw materials and was ideal for the development of functional or intermediate molecules for chemical synthesis, such as glycerol carbonate or glycidol [77].
Maltol is one of the derivatives of biomass, and maltol by-products have a certain synergistic effect with pine chips. Adding less than 10% maltol by-products to pine wood chips to make a fuel blend can improve combustion characteristics and reduce emissions [78]. 1,2-Cyclopentanedione, 3-methyl- is an orthocyclodione, which is an important fine chemical intermediate and is widely used in pharmaceutical, chemical and other industries [79]. Phenol, 2-methyl- can be used in organic synthesis, also as a disinfectant and preservative, is an important pharmaceutical intermediate. It is also the main compound in biooils [80]. Creosol is a lignin derivative of biomass and is a high value-added product as a source of renewable assets of great interest to industry [81, 82]. Similarly, the thermal cracking products detected by Py/GC–MS also contain chemical components such as furfural [55-57], dihydroxyacetone [58, 59], and 5-Hydroxymethylfurfural [60,61]. The analysis of pyrolysis products shows that the chemical components in S. holocarpa wood can be well applied in chemical, bioenergy and other fields. At the same time, the test results of Py/GC–MS and GC–MS were also consistent, which further demonstrates the potential of S. holocarpa wood as become lignocellulosic biomass energy.”
Q17. In Figure 6, please improved the figure scale, because the numbers are small.
Answer: We thank the reviewer very much for the additional proposal. We have improved the scale of the figure in the manuscript.
Action: Since the image is too large, please see our revised manuscript.
Q18. Lines238-259. Please add the discussion results obtained..
Answer: We thank the reviewer very much for the additional proposal. We have added a discussion of this section.
Action: “ The results of 1H-NMR detection showed that S. holocarpa wood was rich in chemical components, including acids, ethers, alcohols, esters, aromatics and other organic compounds. Bio-oil is complex mixtures of highly oxygenated organic components, including almost all kinds of oxygenated organic compounds. S. holocarpa wood has the potential to become a green and sustainable energy source. This was consistent with FT–IR, GC–MS, TGA and PY/GC–MS test results. ”

Author Response
Manuscript Number: molecules-2099583
Title: Potential of Staphylea holocarpa wood for renewable bioenergy
Dear respected editor and reviewers, we would like to express our sincere gratitude to you for writing us the following constructive comments on our manuscript. Also, we appreciate very much for your willingness to check and help to improve the overall contents and quality of our manuscript with your precious time. Thank you so much for your comments and advice. We have made our best efforts to revise and improve our manuscript in an effort to acknowledge the reviewers’ comments accordingly. The comments from the reviewers are retyped below in italics, our responses are typed in normal black font, and the modifications done to the manuscript are also shown in red font. Thank you very much.
Comments from Reviewers
Reviewer #3:
Article titled “Explore the potential of Staphylea holocarpa wood for renewable
bioenergy’ submitted for publication in the journal “Molecules” summarizes the
characterization the wood from Staphylea halocarpa with primary focus on application towards bioenergy. The article is well written where methods and results are clearly described. The article may be accepted for publication however authors are recommended to consider following comments towards further improving the scope and clarity of the described work.
Answer: We thank you for your careful review and for given us a possibility to improve the quality of our manuscript.
Major comments
Q1. ) Authors are focused on exploring the potential of the Staphylea halocarpa wood
for the application in bio energy. Authors are advised to provide little more
background on how the bioenergy or “green energy” is produced with woods
from the enriched species. Authors are recommended to include a paragraph on
this aspect with relevant reference (e.g. bioenergy production from other woods
and different methods) in the introduction.
Answer: We thank the reviewer very much for the comments. We have added to the manuscript an introduction to the bioenergy production from wood.
Action: “Bioenergy comes mainly from wood biomass, and wood-based bioenergy development can play a vital role in achieving energy independence, reducing carbon emissions and promoting rural development [9]. The production status of various wood-based bioenergy products will improve the prospects for the development of wood-based bioenergy. With the advent of advanced afforestation treatments and efficient biotechnology, wood-based bioenergy development can meet the needs of sustainable energy production [10]. About 11% of the world's primary energy consumption comes from biomass. However, ongoing material shortages point to the need to find suitable wood for bioenergy. For example, Eucalyptus pellita and Hevea brasiliensis clone potential wood for bioenergy have net calorific values of 16,502 kJ kg−1 and 19,757 kJ kg−1 [11]. When Tachigali vulgaris was planted at a spacing greater than 6 m2 and a net calorific value above 7.95 MJ/kg, it has a medium basic density and is suitable for bioenergy [12]. Fort McPherson, Northwest Territories, Canada, found that using native willow fragments for 2–37 years can yield greenhouse gas benefits within 0–20 years [13]. Pinus spp. and Quercus spp. were pressed into solid biofuel briquettes that can be used to meet the need to generate low-power heat for the residential sector. The preparation of briquettes from biomass residues is efficient, economical and easy to manufacture and use [14].”
Q2. Authors have done a great job in the characterization of the Staphylea halocarpa
wood. However it is not clear what is the good (practical/meaningful) levels of
key components. Authors are advised to provide some reference levels from other
resources (e.g. woods from other species) or talk about other species which are
considered as standard for bioenergy applications
Answer: We thank the reviewer very much for the comments. We have analyzed the components of organic compounds identified by GC–MS and Py/GC–MS that can be used in bioenergy in the manuscript.
Action: “3.2. GC–MS Analysis
In addition, many chemical components that were conducive to the development of biomass energy were detected in the samples of the four extracts. For example, furfural was an important biomass-derived platform molecule that can be used to synthesize a variety of value-added chemicals. Furfural and its derivatives were promising alternatives to traditional petrochemicals [55]. The furfural industry was constantly evolving. Recently, the annual global production of furfural exceeded 300,000 tons, of which about 70% was produced in China [56]. Furfural and its derivatives were widely used in industrial production in organic solvents, pharmaceuticals, agricultural chemicals, biofuels or fuel additives [57]. One of the most important value-added products obtained from glycerin was dihydroxyacetone. Dihydroxyacetone can also be used as a building block in organic synthesis and is a promising area for the development of novel polymer biomaterials. For example, the design of injectable synthetic biodegradable polymer biomaterials composed of polyethylene glycol and a polycarbonate of dihydroxyacetone [58, 59]. Among the various biomass-derived chemicals, 5-Hydroxymethylfurfural has received great attention due to its potential applications and was listed by the U.S. Department of energy as one of the promising platform chemicals [60]. 5-Hydroxymethylfurfural is a high-value central platform chemical that can be obtained directly from hexose dehydration. The unique structure of 5-Hydroxymethylfurfural gives it high chemical activity and allows it to be transformed through various catalytic processes such as oxidation, hydrogenation and amination. It can be used in the production of high value-added chemicals and liquid fuels such as 2,5-furandialdehyde, 2,5-furandicarboxylic acid, levulinic acid, etc. [61]. In general, the chemical components identified from GC–MS have very good applications in terms of biomedicine and bioenergy. Thus illustrating S. Holocarpa wood has potential as a bioenergy material.
3.4. Py/GC–MS Analysis
Many of the pyrolysis products identified by PY/GC–MS detection can be used in the chemical industry as green energy. For example, acetaldehyde belongs to biomass-derived oxygenated compounds, which are one of the main components of bio-oil. Acetaldehyde is mainly used as a reducing agent and is industrially used in the manufacture of polyacetaldehyde, acetic acid, synthetic rubber, etc. [73]. Formic acid is a major product of carbohydrates derived from biomass and is receiving increasing attention as a sustainable hydrogen source. Formic acid-mediated biomass feedstock can be converted into value-added products, including biofuels, levulinic acid, etc. [74]. Catechol is an industrially relevant chemical with countless applications. It is the most representative basic structure unit in lignin, and it is also the main reaction intermediate and product in biomass or lignin pyrolysis [75]. Catechol plays an important role in many systems by interacting with organic and inorganic compounds. In addition, Catechol crosslinked polymer networks exhibit remarkable mechanical strength, good adhesion and realistic properties [76]. Biomass provides an important source of raw materials and is ideal for the development of functional or intermediate molecules for chemical synthesis, such as glycerol carbonate or glycidol [77].
Maltol is one of the derivatives of biomass, and maltol by-products have a certain synergistic effect with pine chips. Adding less than 10% maltol by-products to pine wood chips to make a fuel blend can improve combustion characteristics and reduce emissions [78]. 1,2-Cyclopentanedione, 3-methyl- is an orthocyclodione, which is an important fine chemical intermediate and is widely used in pharmaceutical, chemical and other industries [79]. Phenol, 2-methyl- can be used in organic synthesis, also as a disinfectant and preservative, is an important pharmaceutical intermediate. It is also the main compound in biooils [80]. Creosol is a lignin derivative of biomass and is a high value-added product as a source of renewable assets of great interest to industry [81, 82]. Similarly, the thermal cracking products detected by PY/GC-MS also contain chemical components such as furfural [55-57], dihydroxyacetone [58, 59], and 5-Hydroxymethylfurfural [60,61]. The analysis of pyrolysis products shows that the chemical components in S. holocarpa wood can be well applied in chemical, bioenergy and other fields. At the same time, the test results of Py/GC–MS and GC–MS are also consistent, which further demonstrates the potential of S. holocarpa wood as become lignocellulosic biomass energy.”
Q3. Authors have primarily focused on characterization of Staphylea halocarpa wood
for the application bio energy. Is it necessary to focus on bioenergy in this article
(or article title) as authors have talked about other application of the different
valued chemicals found in Staphylea halocarpa wood?
Answer: We thank the reviewer very much for the additional proposal. We have added content to the manuscript about the use of organic chemistry for bioenergy, which is in line with our title and overall idea.
Action: “Bioenergy comes mainly from wood biomass, and wood-based bioenergy development can play a vital role in achieving energy independence, reducing carbon emissions and promoting rural development [9]. The production status of various wood-based bioenergy products will improve the prospects for the development of wood-based bioenergy. With the advent of advanced afforestation treatments and efficient biotechnology, wood-based bioenergy development can meet the needs of sustainable energy production [10]. About 11% of the world's primary energy consumption comes from biomass. However, ongoing material shortages point to the need to find suitable wood for bioenergy......
In addition, many chemical components that are conducive to the development of biomass energy were detected in the samples of the four extracts. For example, furfural is an important biomass-derived platform molecule that can be used to synthesize a variety of value-added chemicals. Furfural and its derivatives are promising alternatives to traditional petrochemicals [55]. The furfural industry is constantly evolving. Recently, the annual global production of furfural exceeded 300,000 tons, of which about 70% was produced in China [56]. Furfural and its derivatives are widely used in industrial production in organic solvents, pharmaceuticals, agricultural chemicals, biofuels or fuel additives [57]. One of the most important value-added products obtained from glycerin is dihydroxyacetone. Dihydroxyacetone can also be used as a building block in organic synthesis and is a promising area for the development of novel polymer biomaterials.For example, the design of injectable synthetic biodegradable polymer biomaterials composed of polyethylene glycol and a polycarbonate of dihydroxyacetone [58, 59]. Among the various biomass-derived chemicals, 5-Hydroxymethylfurfural has received great attention due to its potential applications and is listed by the U.S. Department of Energy as one of the promising platform chemicals [60]. 5-Hydroxymethylfurfural is a high-value central platform chemical that can be obtained directly from hexose dehydration. The unique structure of 5-Hydroxymethylfurfural gives it high chemical activity and allows it to be transformed through various catalytic processes such as oxidation, hydrogenation and amination. It can be used in the production of high value-added chemicals and liquid fuels such as 2,5-furandialdehyde, 2,5-furandicarboxylic acid, levulinic acid, etc. [61]. In general, the chemical components identified from GC–MS have very good applications in terms of biomedicine and bioenergy. Thus illustrating S. Holocarpa Wood has potential as a bioenergy material.”
Once again, the authors are thankful to the Editor and Reviewers for providing us valuable feedback/suggestions on the manuscript to improve. We have thoroughly and carefully revised the relevant sections in the manuscript in accordance with the reviewers’suggestions. We hope that the reviewers will be satisfied with the updated version of the manuscript.
Best regards,
The authors

Round 2
Reviewer 1 Report
The revised version of the manuscript looks good, but there are several English language errors in the revised manuscript. It appears that an English Editor did not look over the manuscript. For example, in lines 71 and 91, the revised sentences are still not correct.
Line 71: Cellulose hemicellulose and lignin
This phrase should include a "comma" after the word "Cellulose".
Line 91: As little has been reported about S. holocarpa wood, especially in the field of bioenergy.
This line 91 is still not reading well.
Please have an English Editor review your manuscript and correct for all English errors before publication.
Author Response
Manuscript Number: molecules-2099583
Title: Potential of Staphylea holocarpa wood for renewable bioenergy
Dear respected editor and reviewers, we would like to express our sincere gratitude to you for writing us the following constructive comments on our manuscript. Also, we appreciate very much for your willingness to check and help to improve the overall contents and quality of our manuscript with your precious time. Thank you so much for your comments and advice. We have made our best efforts to revise and improve our manuscript in an effort to acknowledge the reviewers’ comments accordingly. The comments from the reviewers are retyped below in italics, our responses are typed in normal black font, and the modifications done to the manuscript are also shown in blue font. Thank you very much.
Comments from Reviewers
Reviewer #1:
The revised version of the manuscript looks good, but there are several English language errors in the revised manuscript. It appears that an English Editor did not look over the manuscript. For example, in lines 71 and 91, the revised sentences are still not correct.
Q1. Line 71: Cellulose hemicellulose and lignin
This phrase should include a "comma" after the word "Cellulose".
Answer: We thank the reviewer very much for the comments. We have made the changes in the manuscript.
Action: “ Cellulose, hemicellulose and lignin make up lignocellulosic biomass, ”
Q2. Line 91: As little has been reported about S. holocarpa wood, especially in the field of bioenergy.
This line 91 is still not reading well.
Answer: We thank the reviewer very much for the comments. We have made the changes in the manuscript.
Action: “ To the best of author’s knowledge, limited research has been reported about S. holocarpa wood, especially in the field of bioenergy.”
Q3. Please have an English Editor review your manuscript and correct for all English errors before publication.
Answer: We thank the reviewer very much for the comments. We have invited the English Editor to make grammatical corrections to our manuscript. And traces of modification are retained in the manuscript, please see our manuscript.
Once again, the authors are thankful to the Editor and Reviewers for providing us valuable feedback/suggestions on the manuscript to improve. We have thoroughly and carefully revised the relevant sections in the manuscript in accordance with the reviewers’suggestions. We hope that the reviewers will be satisfied with the updated version of the manuscript.
Best regards,
The authors